# Understanding vegetable farmers' adoption, dis-adoption, and non-adoption decisions of pest management by pheromone trapping

Muhammad Humayun Kabir[1], Sk. Md. Nur-e-Alam[1], Avishek Datta[2], Mou Leong Tan[3], Md. Sadique Rahman[4]*

1 Department of Agricultural Extension and Information System, Sher-e-Bangla Agricultural University, Dhaka, Bangladesh, 2 Agricultural Systems and Engineering, Department of Food, Agriculture and Bioresources, School of Environment, Resources and Development, Asian Institute of Technology, Klong Luang, Pathum Thani, Thailand, 3 Section Geography, Universiti Sains Malaysia, George Town, Malaysia, 4 Department of Management and Finance, Sher-e-Bangla Agricultural University, Dhaka, Bangladesh

* saadrhmn@yahoo.com

**Data Availability Statement:** All relevant data are within the paper and its Supporting information files.

## Abstract

The use of pheromone traps can minimize the excess application of synthetic insecticides, while can also benefit the environment. The use of pheromone traps has been promoted and suggested to vegetable farmers of Bangladesh for widespread adoption. However, the majority of farmers have continued to spray insecticides instead of using pheromone traps. The present study investigated the factors influencing farmers' adoption, dis-adoption, and non-adoption behavior of pheromone traps for managing insect pests. Primary data were collected from 438 vegetable growers. Data were analyzed using descriptive statistics and multinomial logistic regression. About 27% of the farmers abandoned the technique shortly after it was adopted as it was time-consuming to manage insect pests. Marginal effect analysis revealed that the likelihood of continued adoption was 34.6% higher for farmers who perceived that pheromone traps were useful in controlling insect pests. In contrast, the likelihood of dis-adoption was 16.5% and 10.4% higher for farmers who maintained communication with private pesticide company agents and neighbor farmers, respectively. Extension services by government extension personnel might be encouraged and maintained as a key component in increasing farmer awareness regarding the use of pheromone trap. Strategies to promote pheromone traps in vegetable production should highlight the positive impacts to farmers and the environment, as this would most likely lead to their continued and widespread use after initial adoption.

## Introduction

The world population is expected to reach 9 billion by 2050, making food availability increasingly important to feed this growing population [1]. Pesticides, an economic and efficient tool for pest management, can help substantially reduce crop yield losses caused by pests and feed the world's growing population [2–6]. However, excess use of pesticides causes detrimental effects on human health and the environment as well as raises the cost of crop production [7–10].

**Funding:** This research was funded by the Ministry of Science and Technology, Government of the People's Republic of Bangladesh.

**Competing interests:** The authors have declared that no competing interests exist.

Bangladesh's crop sector is dominated by rice (*Oryza sativa* L.), although vegetable production surged by 35.3% during the past five years, increasing farm incomes and improving diets [10–13]. However, insect pests pose a severe threat to vegetable production, and farmers heavily rely on insecticides to control them [9, 10, 14, 15]. In Bangladesh, vegetable production uses more quantity of harmful pesticides than cereal production. Vegetable farmers in Bangladesh have been reported to apply significantly higher amount of pesticides (up to six times) compared with rice farming [16]. This excess use of pesticides has forced other countries to consider restricting vegetable imports from Bangladesh [6]. Therefore, the Government of Bangladesh has encouraged the use of integrated pest management (IPM) and announced a national IPM policy [17]. Integrated pest management has the potential to mitigate the negative effects of pesticide by preventing its overuse, thereby assuring environmental sustainability [18].

Integrated pest management is a central organizing principle that guides pesticide use, emphasizes non-chemical or eco-friendly pest control measures, and recommends synthetic pesticide as the last resort [19]. Integrated pest management is a combination of numerous techniques, one of which is sex pheromone-based mass trapping of male insects. Pheromones are naturally occurring substances or chemical compositions that cause another member of the same species to respond behaviorally. Pheromones have been demonstrated to be effective for mass trapping and disrupting the mating of specific insect pests. Pheromone trapping, either alone or in combination with bio-rational spraying, has been reported to conserve natural enemies and control various insect pests, including eggplant fruit and shoot borer (*Leucinodes orbonalis* Guenee), and cotton bollworm (*Helicoverpa armigera*) in Bangladesh [20–25]. The use of pheromone traps is also viewed as a viable option for controlling insect pests and reducing the use of synthetic insecticides in worldwide vegetable production [15, 18, 22, 23, 25]. Studies have suggested that the dependency on insecticides can be reduced by 35% to 50% without reducing the yield of the product by adopting pheromone traps [26, 27]. In Bangladesh, the rate of return on the Bangladesh Agricultural Research Institute's investment in pheromone research varies from 140% to 165% [28]. Millions of dollars have been spent in developing countries like Bangladesh on research, training, and dissemination of environmentally beneficial methods expecting that farmers would accept and continue to use them [29]. However, despite nearly two decades of work to promote pheromone traps for use in farmers' fields, there is still no substantial progress in its adoption on a large scale [30–34].

The adoption of any environmentally beneficial technology, such as pheromone traps, can contingent upon the availability of extension or financial support [35, 36]. Farmers tend to adopt a new technology as long as support is available, and they discontinue using the technology when the support ends. In addition, several socioeconomic, biophysical, institutional, and information constraints can have a significant impact on adoption and dis-adoption decisions [15, 35]. For sustaining the adoption of pheromone traps, the factors for continuous adoption and dis-adoption need to be identified. Research and extension services would have little influence if the factors have not been thoroughly identified. Furthermore, extension initiatives would necessitate more time and resources in order to reach to the farmers.

There is an abundance of literature on the factors influencing the adoption of overall IPM approach [3, 15, 19, 36–40]. However, research on certain IPM techniques, particularly pheromone traps, is limited. Furthermore, previous studies have concentrated on the adoption or non-adoption of the overall IPM approach, while ignoring the dis-adoption issue. Mass trapping of male insects with pheromones can significantly reduce pesticide use and benefit the environment. Unfortunately, many farmers who initially adopted the technique abandoned it later. To make pheromone traps suitable for adoption by farmers, the underlying causes of continued adoption and dis-adoption need to be explored. This has not been thoroughly

investigated until now. Therefore, the present study was conducted to identify the factors influencing farmers' decisions on the adoption, dis-adoption, and non-adoption of pheromone traps for insect pest management.

## Materials and methods

### Study areas

The study was carried out in Jashore, Bogura, and Narsingdi districts (administrative unit) of Bangladesh. These districts were purposively selected due to an intense vegetable production in these areas [41]. The focus was on vegetable farming as vegetables are an important component of the cropping pattern and provide higher profit [10, 42]. The locations were also selected in part due to the emphasis placed by public agricultural extension service providers on providing technical and financial assistance to farmers for increasing the use of pheromone traps to manage insect pests in their vegetable farms.

### Sampling technique

The study followed multistage sampling procedure to select the respondents. At first, three districts were selected. Secondly, three subdistricts, one from each district, were selected based on vegetable production and pheromone trap program diffusion. Third, from each subdistrict, four villages with at least some farmers using pheromone traps for more than three years were selected. Finally, a list of total vegetable farmers from the selected 12 villages was prepared with the assistance of subdistrict agricultural extension officers. The population of the study was 2,192 vegetable growers from the selected villages. Yamane's formula (equation 1) for determining the sample size was used to determine the minimum sample size for this study [43]. Other researchers have also frequently used this formula for determining the sample size in their studies [44, 45].

$$n = \frac{N}{1 + N(e^2)} \tag{1}$$

where $n$ is the minimum sample size, $N$ is the number of population, and $e$ is the precision level, which was set at 5% (0.05). Eq (1) provided the minimum sample size of 338. However, several other studies have suggested that a sample of 20% of the population could be regarded as a representative of population [46, 47]. As a result, 438 vegetable farmers, taking 146 from each district, were randomly selected and surveyed. After the survey, the selected farmers were divided into three categories: adopters, dis-adopters, and non-adopters of pheromone traps. Farmers who used pheromone traps in the past and are still using them for vegetable insect pest management were considered adopters. Farmers who previously used pheromone traps for vegetable insect pest control but stopped using them once institutional support ended were classified as dis-adopters. Farmers who did not use pheromone traps to manage vegetable insect pests and instead relied on synthetic pesticides were classified as non-adopters.

### Data collection

Face-to-face interviews using structured interview schedule were used to collect the necessary data. Three enumerators were hired and trained for data collection. The interview was conducted in local language to maximize the correctness and authenticity of the data. The interview schedule had three sections. The first section covered socioeconomic characteristics of farmers, such as age, education level, household size, annual (on-farm and off-farm) income, as well as farm characteristics, such as total cultivable area, time spent in vegetable farming,

number of vegetables grown, and household labor size. The second section covered data related to information sources, such as contact with private pesticide company agents, government extension agents, and neighbor farmers, as well as data related to farmers' awareness of pesticide and pheromone trap, such as perception of pesticide use and perception of pheromone trap use. The third section discussed farmers' status in terms of pheromone trap adoption (adoption/dis-adoption/non-adoption).

Four model farmers and two local extension personnel with substantial hands-on experience with pheromone traps reviewed the interview schedule. This phase aided in the refinement of the study's questions, such as improved content definition using appropriate terminology. Furthermore, a pilot survey was conducted with 24 farmers. The information from the pilot survey was used to modify the interview schedule, which was then used in the final survey.

### Ethical consideration

The Review Committee of the Department of Agricultural Extension and Information System, Sher-e-Bangla Agricultural University, Dhaka, Bangladesh, granted ethical approval. Each respondent provided verbal informed consent after being informed of the study's objectives, and the variety of information required. Participation in the study by respondents was voluntary. Respondents were free to refuse or discontinue the interview at any time. If a respondent refused to be interviewed, another household was contacted.

### Variables

In general, the adoption and dis-adoption of any technology can be influenced by several factors, such as farmer's characteristics, farm characteristics, information sources, economic factors, and institutional factors [40, 48]. The selection of the independent variables for this study was based on the priori expectations and previous literature [3, 6, 10, 19, 35, 37, 40]. Eight of the 13 variables were continuous variables, while the others were dummy variables. Table 1 describes the independent variables used in the model.

### Data analysis

Both descriptive statistics and econometric model were used to analyze the data. Depending on the nature of dependent variable, binary logistic regression or multinomial logistic (MNL) regression was used by several authors to identify the factors affecting adoption decision [3, 47, 49]. The MNL model examines the probability of the dependent variable taking one of several defined categories, as influenced by different independent variables. The MNL model is strongly recommended for estimating nominal outcomes of unordered dependent variables/categories. The MNL model can also guarantee choice independence, which indicates that the model is unlikely to enable correlation or replacement between them [50]. The MNL model is a more general variation of the binary logistic model that compares categories using various binary logistic regression models [37]. Because the dependent variable in this study was a multi-category variable (adopter = 2, dis-adopter = 1, and non-adopter = 0), MNL model was employed to find out the factors affecting adoption status of pheromone traps. In formulating the MNL model, farmers are assumed to implicitly maximize their expected utility as they make their decision on the adoption of pheromone traps. Let $P_j$ be the probability that the dependent variable takes the value of the $jth$ level, and $P_J$ be the probability that the dependent variable takes the value of the $Jth$ level (j ≠ J), with the $Jth$ group serving as the base category. The model that describes the behavior of farmers regarding their different adoption decisions

**Table 1. Description of independent variables used in the model.**

| Variable | Notation | Explanation | Expected effect on adoption |
|---|---|---|---|
| **Farmers' characteristics** | | | |
| Age (year) | $X_1$ | Age of household head in years | +/− |
| Education level (year) | $X_2$ | Formal schooling year of household head | + |
| Household size (number) | $X_3$ | Total number of members in the household | + |
| Annual income (BDT[a]/year) | $X_4$ | Total annual income from agriculture and non-agriculture sources in thousand BDT | + |
| **Farm characteristics** | | | |
| Total cultivable land (decimal[b]) | $X_5$ | Total cultivable land area in decimal | +/− |
| Time spent in vegetable cultivation (hours/day) | $X_6$ | Number of hours per day spent in the vegetable farm | − |
| Number of vegetables grown (number) | $X_7$ | Total number of vegetables grown by the household in a season | +/− |
| Household labor size (number) | $X_8$ | Number of household members working in the vegetable farm | + |
| **Information sources** | | | |
| Contact with a private company agent (dummy) | $X_9$ | 1 if communicate with a private pesticide company agent, 0 otherwise | − |
| Contact with a government extension agent (dummy) | $X_{10}$ | 1 if communicate with a government agricultural extension officer, 0 otherwise | + |
| Contact with neighbor farmers (dummy) | $X_{11}$ | 1 if communicate with neighbor farmers regarding vegetable farming, 0 otherwise | +/− |
| **Farmers' perceptions** | | | |
| Perception towards pesticide use (dummy) | $X_{12}$ | 1 if perceived that pesticide use is good against vegetable insect pest, 0 otherwise | − |
| Perception towards pheromone trap (dummy) | $X_{13}$ | 1 if perceived that pheromone trap is good against vegetable insect pest, 0 otherwise | + |

Note:

[a] = Bangladeshi currency, 1 USD = 85 BDT;

[b] = 247 decimal makes a hectare.

is expressed in Eq (2):

$$\ln\left(\frac{P_j}{P_J}\right) = \beta_0 + \sum_{i=1}^{13} \beta_i X_i + u_i, j = 0 - 2 \tag{2}$$

where $X$ is the independent variables, $\beta$ is the unknown parameters to be estimated, and $u$ is the error term. The following empirical model was used in the study to estimate the parameters (Eq 3):

$$Y_i = \beta_0 + \beta_1 X_1 + \beta_2 X_2 + \beta_3 X_3 + \beta_4 X_4 + \beta_5 X_5 + \beta_6 X_6 + \beta_7 X_7 + \beta_8 X_8 + \beta_9 X_9 + \beta_{10} X_{10} + \beta_{11} X_{11} + \beta_{12} X_{12} + \beta_{13} X_{13} + u_i \tag{3}$$

where $Y_i$ is the dependent variable, $X_1$ to $X_{13}$ are the independent variables, $\beta_0$ is the intercept of the regression model, $\beta_1$ to $\beta_{13}$ are the parameters to be estimated, and $u_i$ is the error term.

## Results

### Descriptive statistics

The average age of the respondents was around 50 years. Non-adopters had a higher average year of schooling (5.07 years) than the other two categories (adopters and dis-adopters), albeit the differences were statistically nonsignificant (Table 2). In terms of annual household income, there was a large gap among the three categories. Non-adopters' annual household

**Table 2. Descriptive statistics of independent variables.**

| Variable | Adopter | | Dis-adopter | | Non-adopter | | F-test |
|---|---|---|---|---|---|---|---|
| | Mean | SD | Mean | SD | Mean | SD | |
| **Farmers' characteristics** | | | | | | | |
| Age (year) | 50.63 | 9.74 | 50.46 | 9.51 | 48.39 | 10.83 | 2.19 |
| Education level (year) | 4.29 | 4.23 | 4.85 | 4.03 | 5.07 | 4.16 | 1.48 |
| Household size (number) | 5.26 | 1.54 | 5.13 | 1.78 | 5.14 | 1.45 | 0.32 |
| Annual income (thousand BDT[a]/year) | 295.16 | 282.43 | 315.91 | 365.61 | 446.83 | 451.22 | 7.37*** |
| **Farm characteristics** | | | | | | | |
| Total cultivable land (decimal[b]) | 143.29 | 124.15 | 135.47 | 112.69 | 169.93 | 147.04 | 2.59* |
| Time spent in vegetable cultivation (hours/day) | 5.87 | 1.88 | 5.86 | 1.85 | 6.18 | 2.07 | 1.25 |
| Number of vegetables grown (number) | 4.33 | 1.76 | 4.51 | 1.34 | 4.54 | 1.64 | 0.82 |
| Household labor size (number) | 1.67 | 0.79 | 1.65 | 0.74 | 1.60 | 0.81 | 0.29 |
| **Information sources** | | | | | | | |
| Contact with a private company agent (dummy) | 0.38 | 0.49 | 0.64 | 0.48 | 0.52 | 0.50 | 10.81*** |
| Contact with a government extension agent (dummy) | 0.64 | 0.48 | 0.64 | 0.48 | 0.65 | 0.48 | 0.02 |
| Contact with neighbor farmers (dummy) | 0.92 | 0.27 | 0.94 | 0.24 | 0.86 | 0.34 | 2.43* |
| **Farmers' perceptions** | | | | | | | |
| Perception towards pesticide use (dummy) | 0.80 | 0.40 | 0.94 | 0.24 | 0.86 | 0.34 | 5.86*** |
| Perception towards pheromone trap (dummy) | 0.91 | 0.28 | 0.69 | 0.46 | 0.73 | 0.45 | 14.50*** |
| Observations | 188 | | 118 | | 132 | | — |

Note:

[a] = Bangladeshi currency, 1 USD = 85 BDT;

[b] = 247 decimal makes a hectare,

* and *** indicate significance at 10% and 1% levels, respectively; SD indicates standard deviation.

income was approximately 51% higher compared with that of the adopters. Non-adopters had a larger total cultivable land area than that of the adopters. All the three categories of farmers had the similar length of time spent in vegetable farming, number of vegetables cultivated in their farms, and size of the family labor force. Approximately 64% of farmers who stopped using pheromone traps had established a contact with agents from private pesticide companies, but only 38% of farmers who continued using pheromone traps had made a contact with agents from private pesticide companies. For all three categories of farmers, neighbor farmers were the most prevalent source of knowledge. Pesticides were judged to be beneficial for vegetable production by around 94% of farmers who discontinued the use of pheromone traps, whereas pheromone traps were thought to be effective for vegetable insect pest management by approximately 91% of farmers who continued to use them. There were statistically significant variations among three categories of farmers regarding total annual household income, total cultivable land, interaction with private company agents, contact with neighbor farmers, perception towards pesticide use, and perception towards pheromone trap use.

## Adoption status

The results of adoption status indicate that approximately 43% of respondents adopted and continued using pheromone traps in their vegetable fields, whereas 27% of respondents adopted pheromone traps in the past but discontinued using this technique. Approximately 30% of respondents did not adopt pheromone traps (Table 3).

**Table 3. Adoption status of pheromone trap among the vegetable growers.**

| Category | Frequency | Percentage |
|---|---|---|
| Adopter | 188 | 42.92 |
| Dis-adopter | 118 | 26.94 |
| Non-adopter | 132 | 30.14 |
| Total | 438 | 100 |

## MNL regression

Table 4 shows the results of the MNL model of the probability of adoption and dis-adoption of the pheromone trap by the farmers. Non-adopter was included as the base category. Statistically significant Wald Chi-square value indicates satisfactory fit of the model. The log-likelihood value and pseudo $R^2$ value imply that the model has a strong explanatory power.

The likelihood of adoption of pheromone traps (as against to the base category of non-adoption) increases with an increase in age and perceptions of farmers towards pheromone traps, while the likelihood of adoption decreases with an increase in annual income and time spent in vegetable farming. A one-year rise in age increases the chance of adoption by 0.3%. Similarly, the likelihood of adoption is 34.6% higher for farmers who perceived that

**Table 4. Factors affecting adoption, dis-adoption, and non-adoption of pheromone trap among the vegetable growers.**

| Variable | Adoption | | | Dis-adoption | | |
|---|---|---|---|---|---|---|
| | Coefficient | Robust standard error | Marginal effect | Coefficient | Robust standard error | Marginal effect |
| **Farmers' characteristics** | | | | | | |
| Age | 0.022* | 0.013 | 0.003 | 0.019 | 0.013 | 0.001 |
| Education level | −0.024 | 0.030 | −0.006 | 0.007 | 0.033 | 0.004 |
| Household size | 0.022 | 0.082 | 0.005 | 0.001 | 0.099 | −0.002 |
| Annual income | −0.002*** | 0.001 | −0.0002 | −0.001 | 0.001 | −0.000 |
| **Farm characteristics** | | | | | | |
| Total cultivable land | 0.003 | 0.002 | 0.0001 | 0.002 | 0.002 | 0.000 |
| Time spent in vegetable cultivation | −0.113* | 0.068 | −0.016 | −0.101 | 0.074 | −0.007 |
| Number of vegetables grown | −0.019 | 0.080 | −0.000 | 0.039 | 0.080 | 0.009 |
| Household labor size | 0.070 | 0.177 | 0.011 | 0.048 | 0.187 | 0.001 |
| **Information sources** | | | | | | |
| Contact with a private company agent | −0.383 | 0.252 | −0.160 | 0.642** | 0.294 | 0.165 |
| Contact with a government extension agent | 0.029 | 0.256 | 0.003 | 0.028 | 0.282 | 0.002 |
| Contact with neighbor farmers | 0.440 | 0.387 | 0.020 | 0.791* | 0.482 | 0.104 |
| **Farmers' perceptions** | | | | | | |
| Perception towards pesticide use | −0.424 | 0.345 | −0.223 | 1.088** | 0.470 | 0.256 |
| Perception towards pheromone trap | 1.382*** | 0.323 | 0.346 | −0.080 | 0.290 | −0.167 |
| Constant | −0.974 | 1.071 | — | −2.353** | 1.156 | — |
| Log pseudolikelihood | −427 | | | | | |
| Pseudo $R^2$ | 0.29 | | | | | |
| Wald Chi-square | 72*** | | | | | |
| Number of observations | 438 | | | | | |
| Base category | Non-adoption | | | | | |

Note:

*, **, and *** indicate significance at 10%, 5%, and 1% levels, respectively.

pheromone traps are useful in controlling insect pest populations. An increase in annual income by BDT 1,000, keeping other variables constant, would decrease the likelihood of adoption by 0.02%. Time spent in vegetable cultivation, on the other hand, has a detrimental impact on adoption. According to the results, every additional one hour of work in vegetable fields reduces the chance of adoption by 1.6%.

The likelihood of dis-adoption (as against to the base category of non-adoption) increases with contact with private pesticides company agents, contact with neighbor farmers, and perceived benefits of pesticides. The chance of dis-adoption is 16.5% and 10.4% higher for farmers who retained communication with private pesticide company agents and neighbor farmers, respectively. According to the marginal effect of farmers' perceptions of pesticides, the likelihood of dis-adoption is 25.6% higher among farmers who believed that the use of pesticides is beneficial for vegetable cultivation.

## Discussion

Descriptive statistics indicated that there is a significant difference in annual income among the three categories of farmers. Non-adopters had higher annual incomes than adopters. Prior researches have indicated that the majority of Bangladeshi farmers, despite the cost advantage of pheromone traps over pesticide application, did not want to adopt environmentally friendly agricultural technologies, such as pheromone traps, due to the time-consuming nature of managing insects [10, 51]. As a result, farmers with higher income purchase synthetic pesticides from the market and apply them to their vegetable fields for fast control of pests.

Descriptive statistics also suggested that non-adopters possess larger area of cultivable land compared with adopters. However, previous studies have suggested that environmentally friendly agricultural technologies, like pheromone traps, are more successful when applied in larger scale [40]. Pheromone trapping could improve pest management outcomes if large farmers could dedicate more land to the practice. However, in Bangladesh, pheromone traps are often not readily available in local markets, and large farmers cannot wait as they require immediate insect pest control measures. As a result, greater attention should be paid to large farmers in terms of making the pheromone traps available and extension services.

According to descriptive statistics, communication with private company extension agents varies significantly among the three categories of farmers. Dis-adopters and non-adopters interact with private company extension agents more frequently than adopters. Since most pheromone traps in Bangladesh are made available by government initiatives, private companies are primarily interested in selling pesticides. Private company extension agents encourage farmers to control insect pests with pesticides. In order to reduce pesticide use, private sector participation in the production and marketing of pheromone traps is required. The findings also indicated that there are considerable disparities among farmer categories in terms of their perceptions of the advantages of pesticides and the usage of pheromone traps. Therefore, raising knowledge and awareness can help ensure the continuing use of environmentally beneficial farming technologies, such as pheromone traps [51, 52].

The adoption analysis revealed that about one-third of the respondents dis-adopted pheromone trap. Farmers were asked about their reasons for dis-adoption. Most of the farmers mentioned rapid action of pesticide in controlling pests, a paucity of pheromone trap materials, and a lack of technical knowledge and information as barriers to the continued use of pheromone trap. The result is consistent with the findings of previous studies, which indicated that pheromone traps are slow to work and more effective when applied to a larger area [40, 51]. Therefore, more field demonstrations and training on the use of pheromone traps are required, which would increase farmers' knowledge and encourage them to adopt the technology.

Adoption analysis further suggested that continued adoption of pheromone traps was influenced by farmer's age, perceptions regarding usefulness of the pheromone traps, yearly income, and time spent in vegetable farming. A positive influence of age indicates that older farmers are more inclined to continue using pheromone traps in vegetable production than younger farmers. This might be because older farmers have more farming experience and are aware of the detrimental consequences of pesticides, which may motivate them to continue using pheromone traps.

The negative annual income coefficient indicates that increased income diminishes the chance of adoption. According to Abdollahzadeh *et al.* [37], adoption of environmentally friendly agricultural technologies reduces as income increases. Farmers with more incomes may invest in agricultural machinery, such as sprayers, which may encourage them to continue using synthetic pesticides. Another possible reason is that pheromone traps are not as readily available in local markets as pesticides. Therefore, when farmers have a higher income, they are more likely to purchase readily-available pesticides. Therefore, increasing farmers' knowledge of the harmful effects of pesticides could improve the current situation.

Farmers who believed that pheromone traps were advantageous for vegetable production were more likely to continue using them. Abdollahzadeh *et al.* [35] also suggested that increasing farmers' perceptions of the negative effects of pesticide application is critical to increasing the adoption of alternative pest management approaches. Disseminating pertinent information that might enhance farmers' knowledge can be critical to continued adoption.

Adoption is negatively impacted by the time spent in vegetable cultivation. Using pheromone traps in farming is time consuming than using synthetic pesticides [51]. Farmers must manually place pheromone traps in their vegetable fields and monitor them on a regular basis, which requires more effort and time in farming. Consequently, this activity might reduce the likelihood of participation in other income-generating activities, which could discourage adoption. Relevant authorities are required to devise and implement specific policies to improve the situation. For instance, if household members, particularly women, are encouraged and trained to install and monitor pheromone traps in the vegetable fields, their counterparts would have ample time to engage them in other income-generating activities.

The findings further suggested that communication with private pesticide company agents and neighbor farmers influenced the dis-adoption of pheromone traps. Integrated pest management has a lower potential for private sector engagement than synthetic pesticides as many of the approaches need improved management rather than commercialized technology [29]. Norton *et al.* [53] also highlighted that the private sector might hinder IPM implementation. Thorburn [54] reported that once government funding was discontinued in Indonesia, pesticide makers and dealers rapidly launched strong advertising campaigns, essentially replacing IPM with chemical restrictions. Therefore, IPM policy should emphasize the inclusion of the private sector in the production and marketing of pheromone traps in order to decrease pesticide use.

The positive association between contact with neighboring farmers and dis-adoption of pheromone traps may be due to the fact that the majority of farmers in Bangladesh use synthetic pesticides in their crop fields and believe that pesticides are a more effective tool for pest management than pheromone traps [1]. Therefore, when a farmer discusses about pheromone traps with them, they may advocate for the use of pesticides. Abdollahzadeh et al. [37] also indicated that farmers' perceptions were crucial elements in technological adoption as they soon resulted in the discontinuation of the technology in farms. This indicates that strategies aimed at promoting pheromone traps in vegetable production should emphasize their positive impacts to farmers to encourage continued and long-term use following initial adoption.

## Conclusions

Using farm-level cross-sectional data collected through face-to-face interviews, this study identifies the factors that influence the adoption, dis-adoption, and non-adoption of phero-mone traps in vegetable farming. Farmers were found to be unenthusiastic in continuing to use pheromone traps in vegetable production, owing to a lack of knowledge, awareness, and the time-consuming nature of controlling insects. Therefore, extension services and training support should be encouraged and maintained as a key component in increasing farmer knowledge and awareness regarding environment-friendly agricultural practices like phero-mone traps. Since the use of pheromone trap is more labor-intensive and time-consuming for farmers, it may be possible to minimize this problem by training and incentivizing farm-er's spouse and relatives to use pheromone traps in vegetable farms. Large-scale farmers should be given special consideration as they are more likely to abandon the usage of phero-mone traps and resort to synthetic insecticides for insect pest management. It is recom-mended that government agricultural extension agencies continue to monitor the adoption process and provide adequate extension services to farmers following initial adoption. To encourage the continued use of pheromone traps, initiatives should be taken to commercial-ize pheromone trap production by bringing in private sector enterprises. This will increase the availability of pheromone materials in the market, aid in widespread adoption, and ulti-mately improve environmental health.

The study is limited to the vegetable farmers of three locations. In order to develop a com-prehensive scenario for the adoption of pheromone traps in Bangladesh, future research may find a large-scale survey useful. Farmers' awareness and knowledge can be increased through a variety of techniques, including field day, demonstration, and the mass media. Future research may consider determining the most cost-effective method of disseminating information regarding environment-friendly agricultural technologies, such as pheromone traps.

## Supporting information

**S1 Data.**
(DTA)

## Acknowledgments

The authors are grateful to the respondents and enumerators for their cooperation during data collection.

## Author Contributions

**Conceptualization:** Muhammad Humayun Kabir, Sk. Md. Nur-e-Alam.

**Data curation:** Muhammad Humayun Kabir, Sk. Md. Nur-e-Alam.

**Formal analysis:** Md. Sadique Rahman.

**Funding acquisition:** Muhammad Humayun Kabir, Sk. Md. Nur-e-Alam.

**Methodology:** Muhammad Humayun Kabir, Md. Sadique Rahman.

**Project administration:** Muhammad Humayun Kabir.

**Supervision:** Avishek Datta.

**Writing – original draft:** Muhammad Humayun Kabir, Mou Leong Tan, Md. Sadique Rahman.

**Writing – review & editing:** Avishek Datta, Mou Leong Tan, Md. Sadique Rahman.

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
