## [Decision Letter · Decision Letter 0]

26 Jul 2023

PONE-D-22-34820Understanding vegetable farmers’ adoption, dis-adoption, and non-adoption decision of pheromone trap: An environment-friendly approach for monitoring insect pestPLOS ONE

Dear Dr. Rahman,

Thank you for submitting your manuscript to PLOS ONE. After careful consideration, we feel that it has merit but does not fully meet PLOS ONE’s publication criteria as it currently stands. Therefore, we invite you to submit a revised version of the manuscript that addresses the points raised during the review process.

Authors to make strong justifications regarding novelty. Strengthen the conclusion. 

We look forward to receiving your revised manuscript.

Kind regards,

Muhammad Khalid Bashir, PhD

Academic Editor

PLOS ONE

Journal Requirements:

2. Our internal editors have looked over your manuscript and determined that it is within the scope of our Sustainability and the Circular Economy Call for Papers. The Collection will encompass a diverse and interdisciplinary set of submissions related to sustainability and the circular economy, focusing on production models, business plans, and the contribution of global initiatives to increased sustainability in economic, environmental, and social terms. Additional information can be found on our announcement page: https://collections.plos.org/call-for-papers/sustainability-and-the-circular-economy/. If you would like your manuscript to be considered for this collection, please let us know in your cover letter and we will ensure that your paper is treated as if you were responding to this call. If you would prefer to remove your manuscript from collection consideration, please specify this in the cover letter.

Reviewers' comments:

Reviewer's Responses to Questions

**Comments to the Author**

1. Is the manuscript technically sound, and do the data support the conclusions?

Reviewer #1: Yes

Reviewer #2: Partly

2. Has the statistical analysis been performed appropriately and rigorously? 

Reviewer #1: Yes

Reviewer #2: Yes

3. Have the authors made all data underlying the findings in their manuscript fully available?

Reviewer #1: Yes

Reviewer #2: No

4. Is the manuscript presented in an intelligible fashion and written in standard English?

Reviewer #1: No

Reviewer #2: No

5. Review Comments to the Author

Reviewer #1: I have read the manuscript with great interest. It is an important piece of work as authors investigate reasons for adoption, disadoption and nonadoption of pheromone trapping for pest control. Their recommendation will benefit Extensions, policy makers and farmers. Indeed, many IPM technologies including mass trapping by pheromone traps are not used by farmers specially in developing countries despite their substantial promotion and training of farmers. So, why this is happening? Authors tried to focus few points of diasadoption and nonadoption in this manuscript. Interesting. However, this very nice piece of research are not well presented, in other words, I felt the manuscript is not well-written. There are lacking of linking (coherency) in paragraph to paragraph and even line to line. I felt justification of the study, novelty of the research and core conclusion is not clear. I suggest authors to read the manuscript line by line carefully and make the story understandable. My section by section comments are provided in a pdf file.

Reviewer #2: The paper is nicely framed to put forth farmers’ perceptions regarding adoption, dis adoption and non-adoption of pheromone trap by farmers in Bangladesh. The abstract portrays interesting insights necessary to lead a brief introduction of the research. Introduction is succinct and coherent. The background presented on different aspect of IPM approaches and methods are aptly described but there is need to give some detail regarding this method (pheromone trap). Title describes the main theme of the manuscript, but objectives of the study should be clearly described preferably at the end of the introduction part.

Differentiate between non-adoption and dis-adoption as mentioned in title?

Overall manuscript needs to be revised thoroughly for linguistic improvement. So, I would suggest having English editing of the manuscript before submitting the revised manuscript.

Also add some evidence of adaptation of pheromone technique in other parts of world

Result section of the manuscript does not have any support from literature to ensure the authenticity of your results. Add such references.

6. PLOS authors have the option to publish the peer review history of their article (what does this mean?). If published, this will include your full peer review and any attached files.

Reviewer #1: No

Reviewer #2: No

---

## [Author Response · Author response to Decision Letter 0]

25 Aug 2023

13 August 2023

Dr. Muhammad Khalid Bashir

Academic Editor

PLOS ONE

Dear Dr. Bashir

Enclosed please find the revised version of our manuscript entitled “Understanding vegetable farmers’ adoption, dis-adoption, and non-adoption decision of pheromone trap: An environment-friendly approach for monitoring insect pest” (Manuscript Number: PONE-D-22-34820). Listed below are our comments and changes in response to the comments from the Editors and Reviewers.

Reviewer #1:

Comment 1: I have read the manuscript with great interest. It is an important piece of work as authors investigate reasons for adoption, disadoption and nonadoption of pheromone trapping for pest control. Their recommendation will benefit Extensions, policy makers and farmers. Indeed, many IPM technologies including mass trapping by pheromone traps are not used by farmers specially in developing countries despite their substantial promotion and training of farmers. So, why this is happening? Authors tried to focus few points of diasadoption and nonadoption in this manuscript. Interesting. However, this very nice piece of research are not well presented, in other words, I felt the manuscript is not well-written. There are lacking of linking (coherency) in paragraph to paragraph and even line to line.

Response 1: We appreciate your comments. Based on your suggestions, we have significantly revised the manuscript to improve the coherence between paragraphs.

Comment 2: I felt justification of the study, novelty of the research and core conclusion is not clear. I suggest authors to read the manuscript line by line carefully and make the story understandable. My section by section comments are provided in a pdf file.

Response 2: We have revised the manuscript based on the pdf file containing your comments. Below are specifics about your comments and responses. We hope that this revised version describes the novelty and conclusion with clarity.

Comment 3: TITLE: I suggest to make it “Understanding vegetable farmers’ adoption, dis-adoption and nonadoption decision of pest management by pheromone trapping”. Later part of the title distract readers. Moreover, pheromone trap is not used for monitoring only rather its suggested to farmers for mass trapping of male insects. This aspect could be mention in the Introduction.

Response 3: We agree with the comment and have revised the title accordingly.

Comment 4: ABSTRACT: Line (L) 31: “for managing insect pests” can be added after traps for more clarity.

Response 4: We have added the suggested text (please see line 35 in the clean version of the revised manuscript).

Comment 5: L 32: Rationale of the study are not in the abstract that could be add by putting a sentence only. Please add something like this “Mass trapping by pheromone traps has been promoted and suggested to vegetable farmers of Bangladesh, however, majority of farmers continued insecticide spraying instead of trapping.” The present study investigated……..

Response 5: We have added the suggested sentence to the ‘Abstract’ section to convey the rationale of the study (please see line 31-34 in the clean version of the revised manuscript).

Comment 6: L36: What makes them to abandon trapping? Please treat the reasons for not continuing the trapping as next sentence.

Response 6: Farmers mainly abandon pheromone traps as it is time-consuming to manage insect pests. We have added this reason in the ‘Abstract’ section (please see line 38 in the clean version of the revised manuscript).

Comment 7: L38: Why and how they perceived it was useful? Please be explicit. What was in the interview? Asking simply “ is it useful or not?”. Also, have these farmers made more contact with extension and technical person rather than pesticide company representatives?

Response 7: Yes, we just asked them if they believe the use of pheromone traps for insect pest control in vegetable cultivation is beneficial. However, we also believe that the ‘Abstract’ section is not the appropriate place for these specifics. This information belongs to the ‘Materials and Methods’ section. The variables are described in Table 1. Additionally, we have explained the findings in the ‘Discussion’ section.

Comment 8: L39: In connection to my previous comment this contrast becomes more logical. Is it?

Response: Yes, we agree and have revised the ‘Abstract’ section accordingly.

Comment 9: L40: Information and communication technology is a very broad term. Please be precise from your study exactly what information and communication gap you sensed for this non adoption and disadoption.

Response 9: We have revised the ‘Abstract’ section and the entire manuscript based on your suggestion, removing a few broad terms, such as information and communication technology. We hope that this revised version is easy to understand.

Comment 10: L41-43: The private sector’s involvement in commercializing………Why do you think so? What have you found from your study? I have given very detailed comment for Abstract (as an example) as I felt core of this paper are not explicit due to lack of cohesion in story telling. However, this manuscript can be much improved with a little thought and effort in writing. I hope authors will take care that.

Response 10: Thank you for your suggestions. We have revised the manuscript and given appropriate explanation in the ‘Discussion’ section.

Comment 11: INTRODUCTION: A lot of sentence reshuffling can make the introduction far better. As such its very hard to understand the rationale of the study. For example, as pheromone trap and its adoption issues are the core of the study, it can be treated in first and second paragraph. I understand authors tried to come down local from global aspect, however, global aspect (population increase, vegetable production issues) should be shorter.

Response 11: We agree with your comments and have accordingly revised the ‘Introduction’ section. We have reduced the text on the global and local aspects of pesticide use and vegetable production (first two paragraphs of the ‘Introduction’ section). We have also added the justification of the study (please see line 100-107 in the clean version of the revised manuscript).

Comment 12: L77-82: AVRDC’s development of eggplant borer trapping and next to it few sentences on spodolure for cabbage and cotton bollworm. It took quite an effort to understand the story. With reference authors convey that pheromone trapping is an effective method of insect control. Its better to treat those sentences when you introduced general aspect of pheromone trapping in L72. For example, tell about pheromone trap and it has been reported to control variety of insects by using trapping (ref) including eggplant shoot and fruit borer in Bangladesh. Then, what happened to this technology? Is it with farmers and not . Then comes adoption issue . Then rationale of your study. L86-90, please treat with AVRDC issue.

Response 12: We have revised the third paragraph of the ‘Introduction’ section to maintain the logical flow of the writing. We have removed the unnecessary sentences to maintain the logical flow (please see line 72-106 in the clean version of the revised manuscript).

Comment 13: L104-109: Authors indicated novelty of their research. Yes, true there are good papers on IPM adoption issues however there are scanty on pheromone trapping. But this is not novelty of your research. Rather please make explicit that mass trapping by pheromone traps is a nice IPM tools worldwide that can reduce pesticide use substantially and benefit environment. Unfortunately, farmers are not using it. To make this adoptable to farmers, core causes of non and dis-adoption need to be investigated. Until now this has not yet been properly investigated. Then highlight in the conclusion that you left few messages for policy makers to work on pheromone trap adoption. Indeed, many IPM tools are abandoned by farmers probably for same causes you have found for pheromone trapping. So I think your study also is an example of that.

Response 13: We have revised the rationale part of the manuscript and have added the sentences as suggested to make the novelty of the study clear. The following sentences have been added to describe the novelty (please see line 99-104 in the clean version of the revised manuscript):

“Previous studies have concentrated on the adoption or non-adoption of the overall IPM approach, while ignoring the dis-adoption issue. Mass trapping of male insects with pheromones can significantly reduce pesticide use and benefit the environment. Unfortunately, many farmers who initially adopted the technique abandoned it later. To make pheromone traps suitable for adoption by farmers, the underlying causes of continued adoption and dis-adoption need to be explored. This has not been thoroughly investigated until now”.

Comment 14: L110-112: Please shift in conclusion.

Response 14: We have deleted this sentence.

Comment 15: MATERIALS AND METHODS: Seems okay with few edit. Please treat ethical consideration in the paragraph of your interview designing.

Response 15: We have moved the ‘Ethical consideration’ section after the subsection titled ‘Data collection’.

Comment 16: What kind of interview it was? Structured questionnaire? Semi structured or informal interviews?

Response 16: We used face-to-face interviews using structured interview schedules. This information has been added to the ‘Data collection’ subsection.

Comment 17: Have you categorized the adopter, disadopter and non adopter before interviewing? Or you have selected farmers those have heard about trapping or experienced using it and while interviewing you have understood which farmers falls in which category?

Response 17: Before the survey, we did not divide farmers into three distinct categories. We randomly selected farmers and then divided them into three categories. This information has been added to the ‘Sampling technique’ subsection.

Comment 18: RESULTS: Is okay. Table 2 foot note b= 247 decimals makes a hectare.

Response 18: Corrected as suggested.

Comment 19: L261-64: Fits in materials and methods.

Response 19: We believe these sentences represent the model’s suitability-related findings. We did not therefore move it to the Materials and Methods section.

Comment 20: DISCUSSION: Everything is there without linking each other. Its better to mention core results and make a discussion on it. For example, those who have high income are non-adopters and they have larger fields. Please discuss why they are non-adopters and how larger fields make them non-adopter? 

Response 20: We have revised the ‘Discussion’ section to establish a connection between the findings and explanation. This revised manuscript explains why income and land area influence non-adoption (please see line 293-306 in the clean version of the revised manuscript).

Comment 21: L308-326: Please make your message clear.

Response 21: We have revised this section to make it clear (please see line 292-317 in the clean version of the revised manuscript).

Comment 22: L327-331: In my view, core of your manuscript. Unfortunately it discussed less.

Response 22: Few additional explanations have been added to represent the reason for non-adoption (please see line 319-326). Our primary objective was to identify the causes of adoption and dis-adoption. Therefore, after presenting the adoption status, we have explained the reasons in the following paragraphs.

Comment 23: Farmers age, income, size of farming influenced the adoption then why don’t you start from socioeconomic causes and then go for labour and time of implementation issues and then source of information, extension training causes and discuss part by part to reach a conclusion. I would suggest start the discussion with summarizing your core results like “Farmers decision on trapping was influenced by age, income, land size….. of farmers). Then slice out one by one in the next few paragraphs. I find your core is labour and time use in trapping, information from neighbours and exposure to pesticide representatives. 

Response 23: We concur and have accordingly revised the ‘Discussion’ section. We have started as “Adoption analysis further suggested that continued adoption of pheromone traps was influenced by farmer’s age, perceptions regarding usefulness of the pheromone traps, yearly income, and time spent in vegetable farming”. We have then explained the findings one by one (please see line 327-355 in the clean version of the revised manuscript).

Comment 24: L366-368: Please be explicit what is “appropriate effort” and why private sector commercialization needed, why not government commercialization?

Response 24: We have revised the sentences and provided the rationale for commercialization in the private sector (please see line 309-313 in the clean version of the revised manuscript).

Comment 25: CONCLUSION: L381-387: Fits better on starting of discussion.

Response 25: We have deleted the sentences to facilitate comprehension.

Comment 26: L387: Time consuming nature: Is it pheromone traps act slower than pesticides or it takes time to install and manage the traps? Pls be explicit with suitable words.

Response 27: We have edited the sentence with suitable words (please see line 379-381 in the clean version of the revised manuscript).

Comment 28: You have concluded farmers should be given incentive, unavialbility of pheromone traps and lure should be solved. Please discuss the unavailability issue in discussion section. Please make more clearer in discussion why large farmers abandon this technique compared to small farmers, afterwards please conclude that large farmers need special attention.

Response 28: As suggested, we have included a few reasons why large farmers are not using it. We have also mentioned the issue of availability in the ‘Discussion’ section. We believe that, following the revision of the ‘Discussion’ section, the conclusion and recommendations now correspond to the findings.

Comment 29: All in all, please pick core message to tell in conclusion and highlight on your recommendation and further research too.

Response 29: In addition to revising the ‘Conclusions’ section to reflect the key findings, we have also incorporated future research directions.

Reviewer #2:

Comment 1: The paper is nicely framed to put forth farmers’ perceptions regarding adoption, dis adoption and non-adoption of pheromone trap by farmers in Bangladesh. The abstract portrays interesting insights necessary to lead a brief introduction of the research.

Response 1: Thank you for your comments.

Comment 2: Introduction is succinct and coherent. The background presented on different aspect of IPM approaches and methods are aptly described but there is need to give some detail regarding this method (pheromone trap).

Response 2: In the third paragraph of the ‘Introduction’ section, we have added a description of pheromone traps (please see line 66-71 in the clean version of the revised manuscript).

Comment 3: Title describes the main theme of the manuscript, but objectives of the study should be clearly described preferably at the end of the introduction part.

Response 3: We have added the objectives at the end of ‘Introduction’ section (please line 104-106 in the clean version of the revised manuscript).

Comment 4: Differentiate between non-adoption and dis-adoption as mentioned in title?

Response 4: The terms ‘non-adoption’ and ‘dis-adoption’ are described in the ‘Sampling technique’ subsection.

Comment 5: Overall manuscript needs to be revised thoroughly for linguistic improvement. So, I would suggest having English editing of the manuscript before submitting the revised manuscript.

Response 5: We have carefully edited the entire manuscript and hope that the language is now clear and simple to comprehend.

Comment 6: Also add some evidence of adaptation of pheromone technique in other parts of world.

Response 6: In the ‘Introduction’ section, we have included several examples of the worldwide use of pheromone traps (please see line 76-79 in the clean version of the revised manuscript).

Comment 7: Result section of the manuscript does not have any support from literature to ensure the authenticity of your results. Add such references.

Response 7: Since we have separate the ‘Results’ and the ‘Discussion’ sections, we have only reported the results in the ‘Results’ section and have explained them in the ‘Discussion’ section by comparing them to other worldwide studies.

Editor comment

Comment 1: Authors to make strong justifications regarding novelty. Strengthen the conclusion.

Response 1: Based on the suggestion of the Reviewers, we have added a few sentences to emphasize the originality of the study (please see line 98-104 in the clean version of the revised manuscript). We have also revised the conclusion so that it corresponds with the findings.

Additional responses

We have modified the title page, headings, subheadings, and references to conform to the journal’s style.

We have changed the data availability statement. All data are now available within the paper.

---

## [Decision Letter · Decision Letter 1]

18 Sep 2023

Understanding vegetable farmers’ adoption, dis-adoption, and non-adoption decisions of pest management by pheromone trapping

PONE-D-22-34820R1

Dear Dr. Rahman,

We’re pleased to inform you that your manuscript has been judged scientifically suitable for publication and will be formally accepted for publication once it meets all outstanding technical requirements.

Kind regards,

Muhammad Khalid Bashir, PhD

Academic Editor

PLOS ONE

Additional Editor Comments (optional):

Reviewers' comments:

Reviewer's Responses to Questions

**Comments to the Author**

1. If the authors have adequately addressed your comments raised in a previous round of review and you feel that this manuscript is now acceptable for publication, you may indicate that here to bypass the “Comments to the Author” section, enter your conflict of interest statement in the “Confidential to Editor” section, and submit your "Accept" recommendation.

Reviewer #1: All comments have been addressed

Reviewer #2: All comments have been addressed

2. Is the manuscript technically sound, and do the data support the conclusions?

Reviewer #1: Yes

Reviewer #2: Yes

3. Has the statistical analysis been performed appropriately and rigorously? 

Reviewer #1: Yes

Reviewer #2: Yes

4. Have the authors made all data underlying the findings in their manuscript fully available?

Reviewer #1: Yes

Reviewer #2: Yes

5. Is the manuscript presented in an intelligible fashion and written in standard English?

Reviewer #1: Yes

Reviewer #2: (No Response)

6. Review Comments to the Author

Reviewer #1: (No Response)

Reviewer #2: (No Response)

7. PLOS authors have the option to publish the peer review history of their article (what does this mean?). If published, this will include your full peer review and any attached files.

Reviewer #1: **Yes: **Dr. Naznin Nahar

Reviewer #2: **Yes: **Dr. Muhammad Amjed Iqbal, Assistant Professor, Institute of Agricultural and Resource Economics, University of Agriculture Faisalabad, Pakistan

---

## [Editor Report · Acceptance letter]

21 Sep 2023

PONE-D-22-34820R1 

Understanding vegetable farmers’ adoption, dis-adoption, and non-adoption decisions of pest management by pheromone trapping 

Dear Dr. Rahman:

I'm pleased to inform you that your manuscript has been deemed suitable for publication in PLOS ONE. Congratulations! Your manuscript is now with our production department. 

Kind regards, 

on behalf of

Dr. Muhammad Khalid Bashir 

Academic Editor

PLOS ONE